## Research Article

sickle cell disease; adolescents and young adults; psychosocial experiences; qualitative research; Kenya

**Corresponding author:**
Yvonne Akinyi Ochieng;
Email: yvonne.ochieng@duke.edu

# "I just want to be normal": Psychosocial experiences of adolescents and young adults with sickle cell disease in Kenya

Yvonne Akinyi Ochieng[1] , Sonali M. Patel[2], Ashita Nazareth[2], Wilter Rono[3], Liz Owino[3], Cyrus Njuguna Githinji[3], Nancy Midiwo[3], Eric Ayaye Kondiek[3], Chelagat Saina[3], Festus Muigai[3], Melanie Bonner[1,4] and Eve Puffer[1]

[1]Psychology & Neuroscience, Duke University, USA; [2]Duke University, USA; [3]Moi Teaching and Referral Hospital, Kenya and [4]Psychiatry and Behavioral Sciences, Duke Medicine, USA

## Abstract

*Background*: Adolescents and young adults with sickle cell disease (SCD) in Kenya experience psychosocial challenges shaped by developmental transitions and social and health system contexts. Limited research has examined differences across adolescence and young adulthood in low-resource settings. *Methods*: We conducted a qualitative study using focus group discussions and thematic analysis to explore psychosocial experiences across three stages: early adolescence (10–14 years), middle adolescence (15–17 years) and late adolescence or young adulthood (18–25 years). Participants included 54 adolescents and young adults with SCD, 18 caregivers and 18 healthcare providers recruited from three healthcare facilities in western Kenya. *Results*: Three themes emerged: (1) emotional and psychological burdens, including fear, uncertainty and identity-related struggles; (2) social challenges, including peer exclusion, family strain and school-related difficulties and (3) healthcare system barriers, including financial hardship, provider-related stigma and limited transition support. Challenges followed a developmental pattern, with younger adolescents emphasizing pain and vulnerability, middle adolescents highlighting social visibility and school participation and older youth focusing on independence and continuity of care. *Conclusion*: Psychosocial needs vary across developmental stages and are shaped by social and health system contexts. Developmentally responsive support, including pain coping, school engagement, and transition services, is needed in low-resource settings.

## Impact Statements

Young people living with sickle cell disease in Kenya often manage pain, school attendance, social relationships and health care responsibilities at the same time. These challenges change as children grow into adolescence and young adulthood, yet support systems rarely reflect these shifts. This study brings attention to how the everyday experiences of adolescents and young adults with sickle cell disease differ across developmental stages, based on perspectives from young people, caregivers and healthcare providers.

By highlighting how psychosocial challenges vary with age, the findings can help inform more age-appropriate approaches to support within health services, schools and families. For example, younger adolescents may benefit from support that focuses on managing pain and understanding their condition, while older adolescents and young adults likely need greater assistance navigating school transitions, employment and the shift from pediatric to adult health care. The study also points to the importance of coordination between health systems and schools in low-resource settings.

More broadly, this work contributes to global mental health efforts by emphasizing the value of developmentally informed approaches to supporting young people living with chronic illness. While grounded in Kenya, the findings are relevant to other low- and middle-income countries seeking to improve mental health, functioning and quality of life for adolescents and young adults.

## Introduction

Sickle cell disease (SCD) is an inherited blood disorder marked by chronic anemia, recurrent pain and multiorgan complications (Sundd et al., 2019). Globally, more than half a million newborns are affected annually, with approximately 70% born in sub-Saharan Africa (GBD 2021 Sickle Cell Disease Collaborators, 2023). In Kenya, an estimated 14,000 children are born with SCD each

year, making it a significant public health concern (Ministry of Health, Kenya, 2023). Advances in medical care have begun to shift SCD from a fatal childhood condition to a chronic illness, with more individuals surviving into adolescence and young adulthood (Quinn et al., 2010; Piel et al., 2013; McGann et al., 2018). This shift has brought greater attention to the long-term developmental and psychosocial needs of young people living with SCD.

Adolescence and young adulthood are periods when psychosocial well-being is closely shaped by developmental change and everyday social contexts, including school and peer relationships (Arnett, 2000). For young people living with SCD, these periods are often marked by recurrent pain, frequent school absences and uncertainty about their health and future. These experiences can disrupt social participation and contribute to emotional distress and reduced quality of life (Bonner et al., 1999; Compas et al., 2012; Poku et al., 2018). Research from high-income countries has consistently documented the psychosocial burden of SCD during adolescence and young adulthood. A review of 15 studies from the United States and United Kingdom found that disease severity and depression were strong predictors of lower health-related quality of life, alongside social stigma and educational disruption (Ojelabi et al., 2017). Similarly, a United States–based study by Wagner et al. (2004) reported that 41.3% of adolescents exhibited elevated social anxiety, with older adolescents (13–17 years) experiencing greater distress due to peer exclusion and school absences. In the United Kingdom, Dyson et al. (2010) reported that over 75% of students with SCD had negative school experiences, including bullying, misinterpretation of symptoms and limited access to school accommodations.

Evidence from SCD-endemic regions highlights similar psychosocial challenges across diverse cultural and social contexts. In sub-Saharan Africa, studies have documented substantial stigma-related burden and emotional distress among adolescents with SCD. In Nigeria, early work documented substantial emotional distress among individuals living with SCD, with nearly half of participants reporting depressive feelings (Anie et al., 2010). More recent research among adolescents found that over 70% reported moderate to high levels of perceived stigmatization, which was associated with poorer health-related quality of life (Adeyemo et al., 2015). In Tanzania, Ghana and Cameroon, adolescents described being teased, bullied or excluded due to visible symptoms such as jaundice or stunted growth, which contributed to low self-esteem and social withdrawal (Munung et al., 2024). Relatedly, studies from Uganda reported that frequent school absences due to pain episodes contributed to academic difficulties and diminished emotional well-being among adolescents with SCD (Kambasu et al., 2019; Tusuubira et al., 2019).

Beyond sub-Saharan Africa, in India, children and adolescents with SCD have reported emotional distress, reduced social functioning and lower school participation compared with their peers (Patel and Pathan, 2005). More recent qualitative work among tribal populations in India has further highlighted how visible symptoms and frequent hospitalizations contribute to social withdrawal and psychosocial burden (Bhat et al., 2023). In Middle Eastern contexts, stigma related to SCD is often embedded within family and marriage expectations, shaping concerns about disclosure, social relationships and future family roles among adolescents and young adults (Pandarakutty et al., 2020; Al Raqaishi et al., 2022).

Healthcare systems also shape psychosocial experiences for adolescents and young adults with SCD, particularly in low-resource settings. Across sub-Saharan Africa, young people with SCD report fragmented care, limited continuity and challenges during the transition from pediatric to adult services, which were described as contributing to emotional distress and uncertainty (McGann et al., 2017; Inusa et al., 2020). These challenges are often heightened during adolescence and young adulthood, when individuals are expected to assume greater responsibility for managing their health within systems that may offer limited developmentally appropriate care.

Despite this growing literature, important gaps remain. Most studies from sub-Saharan Africa and other SCD-endemic regions treat adolescents and young adults as a homogeneous group, with limited attention to how psychosocial experiences differ across developmental stages such as early adolescence, middle adolescence and young adulthood. In addition, much of the existing research focuses on a single stakeholder perspective, typically the adolescent or young adult, without integrating insights from caregivers and healthcare providers. In Kenya specifically, qualitative research examining psychosocial experiences across developmental stages and social contexts remains limited.

To address these gaps, this study draws on Developmental Systems Theory (DST) and the Socio-Ecological Model (SEM). DST emphasizes that development unfolds through dynamic interactions between individuals and their environments over time, with adolescence and young adults marked by distinct cognitive, social and identity-related changes (Ford and Lerner, 1992). SEM complements this perspective by situating psychosocial experiences within interacting levels of influence, including individual factors, interpersonal relationships and institutional contexts such as schools and healthcare systems (Reupert, 2017). Collectively, these frameworks guided an integrated examination of how psychosocial experiences among adolescents and young adults with SCD vary across developmental stages while being shaped by social and structural contexts. This approach also supported the inclusion of multiple stakeholder perspectives to capture the complexity of psychosocial experiences in Kenya.

## Study aims

This study aimed to (1) explore psychosocial experiences across early (10–14 years), middle (15–17 years) and late adolescence/young adulthood (18–25 years) in Kenya and (2) integrate perspectives of adolescents and young adults, caregivers and healthcare providers to examine how psychosocial experiences are shaped across social, educational and healthcare contexts.

## Method

### Study design

We conducted a qualitative study informed by Community-Based Participatory Research (CBPR) principles. CBPR emphasizes equitable collaboration between researchers and community partners throughout the research process (Israel et al., 1998). This approach was selected to ensure that the study remained contextually grounded and responsive to the lived experiences of adolescents and young adults with SCD, their caregivers and healthcare providers. CBPR informed the study design, development of the focus group discussion (FGD) guides, participant recruitment and interpretation of findings through ongoing collaboration with Kenyan healthcare providers and community stakeholders. FGDs were selected as the primary data-collection method due to their capacity to facilitate peer-driven dialog, reflection on shared experiences and collective insight into psychosocial and health challenges (Nyumba et al., 2018).

## Settings

The study was conducted in western Kenya where an estimated 4.5% of children are born with SCD, and approximately 18% of the population carries the sickle cell trait (Wanjiku et al., 2019; Ministry of Health, Kenya, 2023). In partnership with the Academic Model Providing Access to Healthcare (AMPATH), data collection occurred at three healthcare facilities representing different levels of care within the Western Kenya regional health system. Participating sites included: (1) Moi Teaching and Referral Hospital (MTRH), a tertiary-level referral center in Eldoret offering comprehensive SCD services, including genetic counseling, specialized clinics, multidisciplinary psychosocial support and a range of therapies such as hydroxyurea, and regular blood transfusions. (2) Webuye County Hospital, a regional facility providing routine SCD management. However, it lacks other specialized services. (3) Homabay County Hospital, a county-level facility providing basic diagnostic and treatment services and relies heavily on referrals for complex care. This stratification enabled recruitment across diverse clinical contexts and allowed examination of how variation in healthcare infrastructure shaped adolescents' and young adults' psychosocial experiences and access to care.

## Participants and recruitment

Using purposive sampling (Palinkas et al., 2015), 90 participants were recruited across three stakeholder groups. Sample size was guided by qualitative methodological recommendations suggesting 6–8 FGDs per population subgroup to achieve saturation (Guest et al., 2017). In this study, subgroups were defined by developmental stage and stakeholder type. Saturation was assessed iteratively during coding and was reached when no new themes emerged within or across adolescent age groups and stakeholder FGDs in the final rounds of analysis (Guest et al., 2006).

## Eligibility criteria

Adolescent and young adult inclusion criteria included a confirmed diagnosis of SCD, age 10–25 years, receipt of care at a participating facility and ability to provide informed assent (ages 10–17 years) or informed consent (ages 18–25 years). Developmental stratification into early adolescence (10–14 years), mid-adolescence (15–17 years) and young adulthood (18–25 years) was informed by DST and guided sampling to support exploration of age-related differences in psychosocial experiences. Caregivers were eligible if they were primarily responsible for the daily care or medical decision-making of a child or adolescent with SCD. Healthcare providers were eligible if they were directly involved in the clinical or psychosocial care of adolescents and young adults with SCD at a participating facility.

## Recruitment

Recruitment occurred through clinic-based and peer referral pathways. Kenyan providers on the research team identified potential participants during clinic visits. In parallel, individuals with lived experience shared study details through word of mouth and referred interested participants to the research team. For minors (10–17 years), caregivers were initially contacted through healthcare providers and invited to participate alongside their adolescents. Healthcare providers were recruited through colleague-to-colleague outreach, with research team members directly inviting coworkers to ensure representation across diverse clinical and psychosocial roles involved in SCD care.

## Sample characteristics

The adolescent and young adult sample included 54 participants, of whom 59.3% identified as female. Participants were evenly distributed across three developmental stages: 10–14 years ($M_{age}$ = 12.9 years), 15–17 years ($M_{age}$ = 15.8 years) and 18–25 years ($M_{age}$ = 20.4 years). The caregiver sample comprised 18 participants, predominantly female (61.1%), ranging in age from 25 to 63 years ($M_{age}$ = 43.5 years). Caregivers included mothers (55.6%), fathers (33.3%) and other relatives, including an aunt and grandmother (11.1%). The healthcare provider sample included 18 participants, 66.7% of whom identified as female. Providers represented a range of professional roles involved in SCD care, including nurses, child life specialists, social workers, physicians, pharmacists, nutritionists, laboratory technologists and clinical officers (see Appendix A of the Supplementary Material for more characteristics).

## Data collection procedure

Fifteen FGDs were conducted across the three study sites, with approximately six participants per group and durations of 60–90 min. FGDs were organized by stakeholder group and, for adolescents and young adults, stratified by developmental stage. Before the discussions, participants completed a brief sociodemographic questionnaire to assess age, gender, education level and participant role.

Discussions were facilitated by two bilingual Kenyan moderators trained in FGD facilitation. Sessions were conducted in English, Kiswahili, or a combination, depending on participant preference. A semistructured FGD guide with open-ended questions and probes was used to explore experiences across individual, interpersonal, community and systemic levels. Key topics included emotional well-being, coping strategies, social support networks, stigma, healthcare access challenges and treatment barriers (see Appendix B of the Supplementary Material for FGD guides). To encourage participation among younger adolescents, activities including emotion mapping and paired discussions were incorporated (Darbyshire et al., 2005). All sessions were audio-recorded and transcribed verbatim. Kiswahili transcripts were translated into English by bilingual team members. Participants received 1,000 KES (Approximately $7.75 USD) as reimbursement for transportation.

## Analysis

We conducted thematic analysis (Braun and Clarke, 2006) using a collaborative team approach with 7 researchers from Kenya and the United States, including advocates with lived experience. First, we developed a codebook based on the FGD guide (deductive coding). Team members then reviewed all transcripts, wrote data summaries and identified additional inductive codes that were incorporated into the codebook (Fereday and Muir-Cochrane, 2006). The codes were then organized into a hierarchical structure consisting of three levels: high-order, intermediate-order and third-order codes. All 15 transcripts and the codebook were uploaded into NVivo software (Version 14) for coding (QSR International, 2023). To establish consistency, five transcripts representing all participant groups (adolescents and young adults, caregivers and healthcare providers) were initially coded by one team member and independently reviewed by another to identify any missing or misapplied codes.

Discrepancies were resolved between the two coders and the senior author (E.S.P). This iterative review process served as a team-based calibration step to align coding practices and refine the codebook (MacQueen et al., 1998; Guest et al., 2012). The remaining 10 transcripts were then coded and reviewed by Y.A.O., S.P., N.M. and A.M.N. Following the established coding procedures, all transcripts were reviewed by at least one other team member, and discrepancies were discussed and resolved through consensus meetings with the senior author (E.S.P).

Themes were then identified and synthesized across three levels: individual, social and structural. Representative quotes were selected to illustrate findings. Finally, the results were shared with a subset of study participants during the dissemination phase, along with other community stakeholders including teachers, faith leaders and policymakers, as part of member checking to inform interpretation of results (Birt et al., 2016).

### Ethical considerations

The study and its procedures received approval from Duke University Institutional Review Board and Moi University Institutional Ethics Review Committee. All participants provided written informed consent (caregivers/providers) or assent with caregiver consent (adolescents) prior to participation.

### Results

Three overarching themes captured adolescents and young adults lived experiences with SCD in Kenya: emotional and psychological experiences, interpersonal and social experiences and healthcare access and systemic barriers. Each theme comprised multiple subthemes reflecting age-specific patterns across developmental stages. Findings integrate perspectives from adolescent, caregiver and providers.

### Theme 1: emotional and psychological experiences

#### Subtheme 1.1: fear and uncertainty

Participants described fear and uncertainty stemming from the unpredictable nature of pain episodes, with these concerns manifesting differently across developmental stages. Younger adolescents described fearing sudden pain episodes and feeling restricted in daily activities, particularly play and school attendance. Many expressed frustration and helplessness about their inability to anticipate when pain would strike and described. One shared: *"When you have this disease, you do not feel free. There is always fear. You cannot play the way others do because you know, at any time, pain might come."*

Middle adolescents often internalized their distress and were reluctant to disclose their condition, expressing concern about appearing weak or different. As one stated: *"Sometimes I wonder if I will ever be like my friends. They can do anything without worrying about getting sick."* Older adolescents and young adults voiced concerns about their future health, education and long-term survival. One participant reflected: *"I always wonder if I will live long enough to finish school or have a family. Every time I fall sick, it reminds me that my life might not go as planned."*

Caregivers echoed these fears and described unpredictability and the emotional toll of pain episodes: *"This disease has challenges. During the day, the child is just okay, but at around midnight,* *sometimes unexpectedly, the pain comes. The pain they endure is unbearable; sometimes, I just cry with them."*

#### Subtheme 1.2: identity formation and self-perception

Participants described how experiences of pain and stigma influenced self-concept and sense of belonging. Younger adolescents described feeling "normal" when symptom-free, but "different" during pain crises. As one stated: *"I want to be normal like my friends. When I am in pain, I feel very different, but when I'm okay, I forget about the sickness."*

Middle adolescents expressed frustration about feeling self-conscious and some questioned the fairness of having SCD and what it meant for their self-worth. One shared: *"I try to look strong outside, but inside I feel different."*

Older adolescents struggled to reconcile their diagnosis with emerging adult identities, particularly expectations related to independence, relationships and future roles. One participant explained: *"I feel like my identity is tied to this disease. It's hard to imagine a future where SCD doesn't control everything. When people look at me, do they see me or just my sickness?"*

Caregivers expressed concerns about the impact on self-esteem and belonging. Shared: *"My child used to ask me, why was I born like this? I didn't know how to answer. It broke my heart because I could see he just wanted to feel normal."* Healthcare providers similarly observed identity-related struggles and noted that younger adolescents lacked awareness of long-term implications, while older participants actively questioned their place in society.

### Theme 2: interpersonal and social experiences

#### Subtheme 2.1: peer relationships

Peer relationships were described as a source of vulnerability across developmental stages, although the nature of these challenges shifted with age. Younger adolescents reported being excluded from play, often driven by fears that they might become sick, which contributed to loneliness and early awareness of difference. One participant shared: *"Sometimes when you are playing with others, they feel you are going to be sick and they remove you from their game, which makes me lonely."*

During middle adolescence, concerns about peer acceptance intensified. Adolescents often avoided disclosing their condition in order to maintain peer acceptance. They described indirect consequences of stigma, such as teasing, rumors and assumptions about physical weakness. Rather than overt exclusion, stigma was often experienced indirectly through changes in how peers interacted with them. As one participant explained: *"I don't like telling people I have sickle cell because they start treating me like I'm weak. Sometimes it's better if they don't know."*

Older adolescents and young adults emphasized peer relationships in the context of romantic partnerships and future social roles. Participants expressed fears of rejection and concerns about being seen as a burden, particularly in relation to marriage and long-term relationships. One of them reflected: *"I want to get married one day, but sometimes I wonder if anyone will want to be with me knowing I have sickle cell."* Caregivers also reported experiencing stigma within communities, noting that some parents discouraged their children from interacting with those with SCD due to misconceptions about contagion.

#### Subtheme 2.2: family relationships

Participants described how family relationships influenced their experiences of autonomy and dependence across developmental

stages. In early adolescence, close caregiver involvement, particularly from mothers, was generally accepted and framed as necessary for managing illness-related vulnerability. One shared: *"When I am sick, my mother is always there. She tells me not to worry because she will help me."*

During middle adolescence, family dynamics became more complex as adolescents sought greater independence. Protective caregiving practices were increasingly experienced as restrictive, particularly when health-related concerns limited social participation or school activities. Adolescents described frustration when parental decisions constrained their autonomy, as one explained: *"I wanted to go for a school trip, but my parents said no because they were afraid I would get sick. I feel like I miss out on so many things."* Among older adolescents and young adults, family support remained essential but was often accompanied by feelings of guilt and concern about burdening caregivers. Participants described tension between reliance on family assistance and expectations of independence. As one noted: *"I know my family loves me, but sometimes I feel guilty because they have to take care of me so much."* Caregivers and healthcare providers echoed these patterns, noting that families often struggled to balance ongoing care with preparing adolescents and young adults to manage their condition independently.

### Subtheme 2.3: school challenges

School environments were described as critical social spaces where adolescents' health status shaped daily interactions with peers and teachers, as well as academic participation. Across developmental stages, participants described school-based experiences as a source of misunderstanding and exclusion, often driven by limited awareness of SCD among educators. Being labeled as lazy or exaggerating symptoms undermined adolescents' credibility and contributed to emotional distress. One student recalled: *"When I told my teacher I needed to sit in the shade because of my condition, they said I was just being lazy and told me to go back to the sun."*

While these challenges were present across ages, their meaning shifted developmentally. Younger adolescents expressed sadness about missing play and physical education due to illness-related absences. Middle adolescents described heightened embarrassment and peer scrutiny following prolonged absences, with some withdrawing socially. As one participant noted: *"When I miss school because of being sick, I return and find that the teacher has already moved on. I feel lost."*

Older adolescents and young adults emphasized long-term academic consequences, including repeating classes and delayed educational progression, and described experiencing anxiety about future employment and independence. Caregivers and healthcare providers corroborated these experiences and noted inconsistent implementation of medical accommodation across schools and limited institutional support for students with SCD.

## Theme 3: health access and systemic barriers

### Subtheme 3.1: financial barriers and healthcare access

Financial constraints were a persistent barrier to timely and consistent care. Participants described how costs related to transportation, medications and clinic visits disrupted treatment routines, particularly when hospitals were distant or medications were unavailable. Middle and older participants were especially aware of their families' financial struggles and described experiencing guilt and emotional distress. One shared: *"If the medicine runs out and there is no money, I just wait until my parents can buy it again."*

Several participants described minimizing their symptoms to reduce financial burden on caregivers. As one noted: *"I try not to complain too much because I know hospital costs money. I just stay quiet when I feel pain."* Caregivers echoed the emotional toll of these trade-offs. They described having to make impossible decisions between healthcare for their children and other basic needs.

### Subtheme 3.2: provider interactions

Participants described varied experiences with healthcare providers, with interactions shaping their engagement with the health system. In specialized sickle cell clinics, care was generally positive, with providers perceived as knowledgeable and reassuring. One participant explained: *"You'll find the ones who are friendly and will attend to you while reassuring you that you'll be well."*

In contrast, experiences in general healthcare settings were often described as dismissive or uninformed. Participants reported feeling unheard when describing pain and described experiencing frustration and emotional isolation. One shared: *"You tell them you are in pain, and they just look at you and say, 'You'll be fine.'"* Participants also described how systemic challenges, including long waiting times and medication shortages, increased distress and sometimes discouraged continued care. Healthcare providers acknowledged these inconsistencies and noted that while specialized clinics offered comprehensive care, general wards often lacked sufficient training in SCD management.

### Subtheme 3.3: transition to adult care

The transition from pediatric to adult healthcare services was described as abrupt and poorly supported. Participants reported that this shift often occurred without adequate preparation, creating confusion, and disruptions in care during a period of increasing responsibility. In early adolescence, most participants had limited involvement in healthcare decision-making, as caregivers communicated directly with providers. Many described feeling unprepared when expectations of autonomy increased in later years.

Middle adolescents increasingly expressed a desire to engage in discussions about their care but often felt excluded when providers continued to direct conversations toward parents. Older adolescents and young adults described an abrupt and disorienting shift to adult healthcare services. They described sudden expectations to independently manage appointments and treatment without guidance. One participant recalled: *"When I turned 18, everything changed, and no one prepared me for it."* Participants described feelings of abandonment, lapses in care and heightened emotional burden related to this unstructured transition. Healthcare providers echoed these concerns and highlighted the lack of formal transition protocols and limited capacity of adult services to address the developmental needs of young adults with SCD.

Expanded comparisons across developmental stages are summarized in Appendix C of the Supplementary Material.

## Discussion

This study explored the psychosocial experiences of adolescents and young adults with SCD in Kenya across early, middle and late developmental stages by integrating perspectives from youth, caregivers and healthcare providers. Findings indicate that psychosocial challenges are shaped not only by illness burden but also by developmental timing and the social contexts in which adolescents strive to maintain normalcy. Across stakeholder groups, participants emphasized that emotional experiences, relational dynamics

and institutional barriers were described as evolving across adolescence and young adulthood, highlighting the need to understand psychosocial experiences through a developmentally sensitive lens.

Consistent with the DST, adolescents' emotional experiences reflected a developmental progression in how illness-related challenges were perceived and managed. Psychosocial stressors shifted from bodily disruption in early adolescence toward heightened social visibility and identity-related concerns in middle adolescence, and ultimately toward future-oriented anxieties related to independence, education, relationships and long-term survival in young adulthood. These findings align with prior research from high-income settings demonstrating increasing emotional complexity across adolescence among youth with SCD (Ojelabi et al., 2017; Panepinto et al., 2017). However, they extend this developmental framework to a sub-Saharan African context where such evidence remains limited. Treating adolescents with SCD as a homogeneous group risks obscuring critical developmental windows during which psychosocial vulnerabilities and priorities differ substantially.

A socio-ecological perspective further illuminated how institutional contexts amplified or constrained adolescents' psychosocial experiences at different developmental stages. Schools emerged as particularly salient environments in which efforts to maintain normalcy were either supported or undermined. In contrast to settings where formal accommodations for chronic illness are institutionalized, such as the United States (U.S. Department of Education, Office for Civil Rights, 2024), Kenyan schools lacked structured mechanisms to address illness-related absences and physical limitations, leaving academic participation and peer inclusion fragile. These challenges were especially consequential during middle adolescence, when peer belonging and identity formation are developmentally central. Within healthcare settings, the transition from pediatric to adult care represented a structural discontinuity that compounded developmental stressors. Adolescents and young adults described this transition as abrupt and disorienting, while providers highlighted limited system-level preparation to support gradual transfer of responsibility, increasing the risk of disengagement from care. These findings are consistent with global evidence on transition-related vulnerabilities in chronic illness populations (Saulsberry et al., 2019; Shah et al., 2022), while underscoring how resource constraints within Kenya's healthcare system may intensify these risks (Wanjiku et al., 2019).

Integrating perspectives of multiple stakeholders revealed both convergence and productive tension that deepened interpretation of youths' experiences. While caregivers and providers largely corroborated adolescents' accounts of pain burden, emotional distress and school-related challenges, tensions emerged around autonomy and protection. Adolescents emphasized a desire for independence and social normalcy, whereas caregivers expressed heightened protectiveness shaped by fears of medical crises, and providers pointed to systemic constraints that limited individualized support. Rather than reflecting disagreement, these tensions highlight the relational processes through which normalcy is negotiated in chronic illness contexts, situating adolescents' experiences within family and healthcare dynamics alongside individual developmental trajectories.

## Implications

*Clinical implications.* Psychosocial interventions and provider practices should be developmentally tailored. Younger adolescents may benefit from evidence-based pain coping strategies such as cognitive-behavioral techniques and structured peer support to reduce isolation. Middle adolescents require stigma reduction through psychoeducation and guided autonomy-building, including gradual involvement in medication management and appointment scheduling. Older adolescents and young adults need structured psychosocial transition support and vocational counseling to address identity development and adult role transitions.

A feasible transition model in Kenya should embed incremental strategies within existing care structures. Inusa et al. (2020) identified six priorities for transition programming: skills transfer, self-efficacy building, care coordination, knowledge acquisition, linkage to adult services and readiness evaluation. Practical elements may include gradually shifting clinic communication from caregivers to adolescents, designating a provider or nurse to coordinate transition discussions, and where feasible, arranging joint pediatric–adult visits during late adolescence (Nagra et al., 2015; Mulchan et al., 2016).

*School-based implications.* Teacher training programs should address SCD symptoms and their classroom implications, helping educators recognize fatigue or pain as potential crisis indicators and understand that symptoms are often invisible. Evidence from mental health literacy programs in Tanzania and Malawi suggests that brief, structured teacher training can improve knowledge and reduce stigma (Kutcher et al., 2015, 2016); similar approaches could be adapted for chronic physical illness. These supports are particularly critical during middle adolescence. Schools should also implement flexible attendance policies and locally adaptable accommodation plans that specify provisions such as extended assignment time, hydration access and designated rest space during pain episodes. Strengthening collaboration between healthcare providers and schools would help ensure that medical guidance informs classroom practices.

*Policy implications.* Expanding Social Health Insurance Fund coverage to include SCD-specific services such as medication, routine monitoring and psychosocial support would reduce financial barriers that contribute to delayed care and emotional distress. Ghana's inclusion of hydroxyurea in its National Health Insurance Scheme offers a regional model for expanding access to disease-modifying therapy through public–private partnerships (Nyonator et al., 2023). For families in rural areas, transport subsidies or mobile clinic outreach could further improve access to consistent care. Kenya's National Guidelines for Control and Management of Sickle Cell Disease (Ministry of Health, Kenya, 2020) provide a foundation for comprehensive care, although explicit incorporation of psychosocial services and structured transition planning would strengthen their application to adolescent populations. Additionally, integrating SCD awareness into national school health curricula would address stigma at the population level and create more supportive educational environments for affected students.

## Limitations and future research directions

The cross-sectional design limits the ability to capture developmental changes over time. The regional focus on western Kenya may constrain generalizability to other contexts. The focus group format may have influenced disclosure of sensitive experiences due to social desirability bias and group dynamics. Additionally, the sample included primarily caregivers and providers from hospital-based settings, which may not represent all subpopulations, particularly individuals from more remote areas or with limited healthcare access. Future research should include longitudinal studies to examine how psychosocial burdens evolve across adolescence and young adulthood, as well as multisite quantitative studies to assess psychological distress, stigma and healthcare utilization.

## Conclusion

Adolescents and young adults with SCD in Kenya experience significant emotional, social and structural challenges that shift across developmental stages. Addressing these burdens requires coordinated, developmentally responsive approaches integrating psychosocial care, health system strengthening and educational support. By engaging policymakers, healthcare providers and educators, Kenya can move toward a more inclusive and supportive environment for adolescents and young adults with SCD, ultimately improving their quality of life and long-term outcomes.

**Open peer review.** To view the open peer review materials for this article, please visit http://doi.org/10.1017/gmh.2026.10175.

**Supplementary material.** The supplementary material for this article can be found at http://doi.org/10.1017/gmh.2026.10175.

**Data availability statement.** Due to the sensitive nature of this qualitative study and the inclusion of vulnerable adolescent populations, full transcripts and raw data cannot be shared publicly to protect participant confidentiality. De-identified excerpts supporting the findings are available upon reasonable request from the corresponding author, subject to institutional and ethical approvals.

**Acknowledgements.** We thank Webuye County Hospital, Homabay County Hospital and Moi Teaching and Referral Hospital in Kenya for their support and collaboration throughout the study. We are especially grateful to the adolescents and young adults, caregivers and healthcare providers who participated and generously shared their experiences. Special thanks to Shirleen Kadima and Teresa Odero for their dedication to participant recruitment and data collection. We also acknowledge Lydia Rugutt for her support with transcription.

**Author contribution.** Y.A.O. contributed to conceptualization, data collection, methodology development, formal analysis, writing – original draft and writing – review and editing. S.P. contributed to data collection, formal analysis. W.R. contributed to project administration, data collection and coding transcripts. L.O. contributed to conceptualization, data collection and validation. C.S. contributed to data collection and writing, review and editing. E.A. contributed to project administration and data collection. N.M. contributed to project administration and data collection. C.N. contributed to project administration and data collection. M.J.B. contributed to conceptualization, supervision and review and editing of the manuscript. F.N. contributed to conceptualization, supervision, validation and writing – review and editing. E.S.P. contributed to funding acquisition, methodology development, formal analysis, supervision and writing – review and editing.

**Financial support.** This work was supported by the Josiah Charles Trent Memorial Foundation Endowment Fund at Duke University.

**Competing interests.** The authors declare none.

**Ethics statement.** This study was approved by the Duke University Institutional Review Board (Protocol #2024–0016) and the Moi University Institutional Research and Ethics Committee (IREC/2024/884). Written informed consent was obtained from all adult participants. For participants under the age of 18, written informed assent was obtained alongside parental or guardian consent. All procedures complied with international and local ethical guidelines for research involving human participants, including provisions for privacy, voluntary participation and the right to withdraw at any time without penalty.

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
