## [Reviewer Report]

Dear Author,

Thank you for a well though of study. I have one key concern with your study. It is about adolescents? You include young adults to your population of adolescents. What reference did you

---

## [Reviewer Report]

This manuscript presents a well-conducted qualitative study exploring the psychosocial experiences of adolescents with sickle cell disease (SCD) in Kenya across developmental stages. The thematic analysis is detailed, and the integration of perspectives from adolescents, caregivers, and healthcare providers strengthens the contribution. There are, however, several areas where clarity, organization, and focus can be improved for enhancing the clarity and readability.

Comments for revision

Abstract

The abstract is largely descriptive but could better highlight what is novel about this study and explicitly state the methodological approach for clarity.

The abstract states implications in general terms; consider adding a more concrete statement of actionable recommendations that emerge from the study.

Introduction

The Introduction contains extensive text but would benefit from clearer positioning of the study gap specific to Kenya. The rationale should emphasize why developmental differences matter and how existing literature has not addressed these stages in SSA.

The literature review should cover studies from SCD-endemic regions like sub-Saharan Africa, India and the Middle East. These studies should include those on psycho-social impact, and stigma due to SCD. Tightening the Introduction section will increase focus.

The theoretical frameworks are well explained but appear unclear on integrated approach. Explain how DST and SEM, together explains your model in the integrated way. Consider sharpening their relevance to the specific aims of the study.

Methods

Sampling is described as purposive, but the rationale for sample size (54 adolescents + stakeholders) is not explained. Please clarify how you determined the data saturation, especially given the developmental segmentation.

Study design should be more detailed.

In the Setting section, there is extensive detail about facility services. Ensure these descriptions directly connect to how settings shaped recruitment or analysis.

Recruitment involves adolescents, caregivers, and providers, but inclusion and exclusion criteria are not clearly stated. For example, caregivers’ relationship to adolescents and provider roles should be more precisely defined.

In the Data Collection Procedure, the phrase “participants completed a brief socio demographic” is incomplete; specify the details.

Results

The themes are comprehensive and richly supported by quotes; however, the Results section is very long. It may benefit from summarizing some descriptive narrative and using more subheadings or tables to highlight developmental differences.

Many quotes are compelling but could be shortened without losing meaning.

The section on school challenges is strong but lengthy. Consider condensing narrative descriptions and focusing on unique contributions of your data.

Discussion

The Discussion effectively compares findings with global literature, but sometimes too much text is dedicated to re-describing results rather than synthesizing insights.

The section could be more concise by focusing on 3–4 key contributions of the study, and by limiting the repitation of the results.

Some citations may not be necessary or could be grouped thematically.

There is an opportunity to more explicitly address the theoretical frameworks introduced earlier: How specifically did DST and SEM illuminate findings?

The transition-to-adult-care discussion is strong; consider expanding briefly on what a feasible transition model in Kenya might look like.

Implications

This section is informative but can be structured more clearly into clinical, school-based, and policy-level recommendations.

Several recommendations (e.g., psychosocial transition support, evidence-based programs) are broad; consider making them more specific or citing relevant models.

Limitations

The Limitations section is appropriate but could acknowledge potential social desirability bias in FGDs and the influence of group dynamics on disclosure.

Also consider mentioning that caregivers’ and providers’ perspectives may not represent all subpopulations.

---

## [Editor Report]

Dear Ms Yvonne Ochieng.

Your Manuscript: “I Just Want to Be Normal”: Psychosocial Experiences of Adolescents with Sickle Cell Disease in Kenya" has now been reviewed,

---

## [Reviewer Report]

I have gone through the revised version of the manuscript. The manuscript has been improved, as all the reviewers' comments have been addressed.

---

## [Reviewer Report]

Dear Author,

The manuscript is well written and has addressed the comments made by the reviewers and the changes made to the writing style has significantly improved the manuscript.

---

## [Editor Report]

Dear Ms Yvonne Ochieng,

Your revised manuscript ““I Just Want to Be Normal”: Psychosocial Experiences of Adolescents and Young Adults with Sickle Cell Disease in Kenya”, has now been reviewed,